# Pilot Study of Heat-Stabilized Rice Bran Acceptability in Households of Rural Southwest Guatemala and Estimates of Fiber, Protein, and Micro-Nutrient Intakes among Mothers and Children

**DOI:** 10.3390/nu16030460

**Published:** 2024-02-05

**Authors:** Brigitte A. Pfluger, Alexis Giunta, Diva M. Calvimontes, Molly M. Lamb, Roberto Delgado-Zapata, Usha Ramakrishnan, Elizabeth P. Ryan

**Affiliations:** 1Doctoral Program in Nutrition and Health Sciences, Laney Graduate School, Emory University, Atlanta, GA 30322, USA; brigitte.pfluger@emory.edu; 2Department of Food Science and Human Nutrition, Colorado State University, Fort Collins, CO 80523, USA; alexis.giunta23@alumni.colostate.edu; 3Center for Human Development, Fundacion para la Salud Integral de los Guatemaltecos, FUNSALUD, Coatepeque 09020, Quetzaltenango, Guatemala; diva.barrientos@cuanschutz.edu; 4Departament of Pediatrics, Center for Global Health, University of Colorado, Aurora, CO 80045, USA; 5Center for Global Health, Colorado School of Public Health, Aurora, CO 80045, USA; molly.lamb@cuanschutz.edu (M.M.L.); roberto.delgado-zapata@cuanschutz.edu (R.D.-Z.); 6Department of Epidemiology, Colorado School of Public Health, Aurora, CO 80045, USA; 7Department of Community & Behavioral Health, Colorado School of Public Health, Aurora, CO 80045, USA; 8Hubert Department of Global Health, Rollins School of Public Health, Emory University, Atlanta, GA 30322, USA; 9Department of Environmental and Radiological Health Sciences, Colorado State University, Fort Collins, CO 80523, USA; 10Colorado School of Public Health, Colorado State University, Fort Collins, CO 80523, USA

**Keywords:** nutrition, rice bran, dietary intake, micro-nutrients, fiber, Guatemala

## Abstract

Nutrient-dense, acceptable foods are needed in low-resource settings. Rice bran, a global staple byproduct of white rice processing, is rich in amino acids, fibers, and vitamins, when compared to other cereal brans. This pilot study examines the nutritional contribution of rice bran to the daily diets of mother–child pairs in rural southwest Guatemala. Thirty households were screened. Mothers (≥18 years) and children (6 to 24 months) completed 24 h dietary recalls at baseline and after 12 weeks (endline) for diet intake and diversity analyses. During biweekly visits for 12 weeks, households with <5 members received 14 packets containing 60 g of heat-stabilized rice bran, and those with ≥5 members received 28 packets. The macro- and micro-nutrient contributions of rice bran and whole, cooked black beans were included in dietary simulation models with average intakes established between the recalls and for comparison with dietary reference intakes (DRIs). A baseline child food frequency questionnaire was administered. The 27 mothers and 23 children with complete recalls were included in analyses. Daily maternal consumption of 10 g/d of rice bran plus 100 g/d of black beans resulted in all achieving at least 50% of the fiber, protein, magnesium, niacin, potassium, and thiamin DRIs. Daily child consumption of 3 g/d of rice bran plus 10 g/d of black beans resulted in all achieving at least 50% of the magnesium, niacin, phosphorous, and thiamine DRIs. For 15/17 food categories, male children had a higher intake frequency, notably for animal-source foods and coffee. Dietary rice bran coupled with black beans could improve nutritional adequacy, especially for fiber and key micro-nutrients, with broader implications for addressing maternal and child malnutrition in low-resource settings.

## 1. Introduction

Malnutrition results from imbalances either due to nutrient excess (overnutrition) or deficits (undernutrition). Suffering from malnutrition at a young age can have life-long health consequences. The first 1000 days of life that include pregnancy through year 2 of life is a particularly sensitive time when inadequate nutrition can hinder proper development and predispose children to an increased risk of health issues, including chronic disease, in adulthood [1]. Malnourished children face increased vulnerability to and higher risks for morbidity, mortality, and delayed cognitive and mental development that can result in fewer years of schooling and reduced lifetime productivity [2,3,4]. Globally, an estimated 6.7% of children under the age of 5 years are wasted and 22% are stunted, which are two conditions of undernutrition defined by a child’s weight for length and length for age, respectively, and calculated as <−2 SD below the WHO Growth Standards median [5,6]. Children in low-resource settings are disproportionately affected [5]. Guatemala, a Central American country of over 17 million people, has the highest stunting rates in Latin America [5,7]. In Guatemala, an estimated 41.7% of children under two years of age suffer from chronic malnutrition [8,9]. Paradoxically, adult overnutrition is on a rise and with an estimated 32% of women 15–49 years old being overweight or obese, which is defined as having a BMI ≥ 25.0 [8].

White rice, a global staple crop and major calorie source, has low nutrient density [10]. Rice bran, the agricultural co-product of rice processing, is nutrient-rich with unique phytochemicals, amino acids, prebiotic fibers, and vitamins B and E isoforms, when compared to bran from other cereals [11,12]. Many of these bioactive and nutritional components are often lacking in diets in low- and middle-income countries (LMICs), including Guatemala [13]. Our previous research shows that consumption of heat-stabilized rice bran among children 6 to 12 months old can improve weight and length for age z-scores [14,15]. Additionally, rice bran components exhibit anti-oxidant, anti-diarrheal, and anti-inflammatory properties [11,12,16,17,18], which may improve nutritional status by optimizing gut mucosal barrier function [19,20]. Rice bran modulates the gut microbiome and metabolome of children and adults [15,21,22] and has demonstrated protective effects against chronic disease, including colorectal cancer [23,24,25]. Thus, rice bran, if locally developed as a food ingredient, may serve as an affordable, available, accessible, and acceptable food option to help address the double burden of malnutrition; it helps prevent undernutrition while simultaneously providing metabolic support for the control and prevention of chronic diseases that result from overnutrition [11,21].

The objective of this pilot study was to examine the regular diets before dietary intervention of mothers and children in four rural communities in the Retalhuleu Department of southwest Guatemala, and to estimate the nutritional contributions of dietary intervention with heat-stabilized rice bran and/or whole, cooked black beans through simulation models and in comparison with dietary reference intakes (DRIs), or nutrient reference values, established by the Institute of Medicine (IOM) [26].

## 2. Materials and Methods

### 2.1. Study Design and Group

This 12-week prospective study was conducted in four rural communities in the Retalhuleu Department of southwest Guatemala where chronic/acute malnutrition rates are approximately 35% [8]. The study CONSORT diagram shown in Figure 1 highlights the households that consented in-person either via records obtained from the local program Madres Sanas y Creciendo Sanos (Healthy mothers and healthy growth) or snowball sampling. Mothers provided written, informed consent for themselves and their child. Eligibility criteria included mothers being ≥18 years old with a healthy child between 6 and 24 months old. In this study, healthy is defined as having a weight for length z-score (WLZ) ≥−2 SD above the WHO Child Growth Standards [9].

At enrollment (baseline/week 0), household demographic data were collected along with child length and weight. Upon project completion, households received a bag of locally available and consumed food valued at GTQ 100 (~USD 13). All data were collected by a Project Coordinator who is a native Guatemalan Spanish speaker living in the study region. The Project Coordinator was trained prior to study initiation and responsible for all data collected with oversight by our local partners at the Fundacion para la Salud Integral de los Guatemaltecos (FUNSALUD; Retalhuleu Department, Guatemala). Appendix A includes the raw data for all 30 consented households.

### 2.2. Dietary Intervention

Stabl Nutrition^TM^ (St. Louis, MO, USA), formally RiceBran Technologies™ (Scottsdale, AZ, USA), manufactured the food-grade heat-stabilized rice bran (RiBran 300) and analyzed its nutritional content (Appendix A). The rice bran was then packaged into sealed packets/sachets each containing 60 g of rice bran (Western Innovations, Inc., Denver, CO, USA) and shipped to Colorado State University (Fort Collins, CO, USA). The study team transported the rice bran packets to FUNSALUD, upon which they were immediately placed in plastic bins with sealed lids and stored in a dry room. Packets punctured during transportation were discarded.

Beginning at baseline, the heat-stabilized rice bran packets/sachets were distributed to the consented households in a plastic container with a secure lid to help prevent spoilage and deter pests. Biweekly, households with <5 members received 14 packets and were requested to integrate 60 g/day (i.e., one packet) into the household diet. Biweekly, households with ≥5 members received 28 packets and were requested to integrate 120 g/day (i.e., two packets) into the household diet. To assess acceptability, households were given little to no instructions regarding how to integrate rice bran into their daily meals, beverages, and snacks, and had the freedom to incorporate the rice bran into any foods and beverages desired.

### 2.3. Data Collection

#### 2.3.1. Dietary Recalls

Maternal and child nutritional status was estimated before and after incorporating daily heat-stabilized rice bran into household diets. Dietary intakes were determined by 24 h dietary recalls verbally administered at baseline and endline. Mothers provided dietary intake data for themselves and their child that included the meal type (i.e., breakfast, lunch, dinner, or morning/afternoon/evening snack), mealtime, meal location, food type, list of ingredients and quantities for homecooked meals, and amount consumed by the individual. The recalls captured food intake during the day (midnight to midnight) prior to the recall. Maternal and child recalls were analyzed separately. Baseline and endline recalls were first averaged at the individual level and analyzed before dietary intervention (i.e., a regular diet). As the exact amount of rice bran consumed was not known, we simulated the nutritional contributions of different estimated intakes of heat-stabilized rice bran. Generally, cooked black beans are locally available and consumed; we simulated the nutritional contributions of adding the estimated intake of cooked, whole black beans with and without the dietary addition of rice bran. The nutritional compositions of the heat-stabilized RiBran 300 rice bran and whole, cooked black beans used in the simulation models are included in Appendix A.

Responses were recorded on paper and transferred electronically to Nutritionist Pro™ (Axxya Systems, Redmond, WA, USA) to estimate total calorie intake (kcal) and the consumption of approximately 11 macro-nutrients (g) and 19 micro-nutrients (mg or μg). When possible, nutrient data were entered in Nutritionist Pro™ directly from the food packaging. If packaging was not available, the nutritional profile was determined through the databases in Nutritionist Pro™, with priority given to the Food Composition Table for Central America by the El Instituto de Nutrición de Centro América y Panamá (INCAP) and the United States Department of Agriculture (USDA) Standard Reference Database [27]. Appendix A includes a list of the foods reported in the recalls and their data sources. Food sizes and weights were also estimated in Nutritionist Pro^TM^ as grams, milliliters, or item units. In Guatemala, brown sugar is fortified with vitamin A and household salt is fortified with iodine [28]. As the amount of fortification is a range and can vary by brand, vitamin A and iodine were removed from nutrient analysis. Additionally, as fat does not have an established DRI for individuals over 12 months of age, it was also removed from analysis. Vitamin E intake from the distributed rice bran was obtained by adding the amount of alpha-tocopherol plus alpha-tocotrienol isomers from the RiceBran Technologies™ spec sheet (Appendix A).

Nutritional intake from the regular foods recorded in diet recalls (i.e., prior to and without dietary intervention) and those simulated with a rice bran and/or black bean dietary intervention were compared to the DRIs established by the IOM [26]. Nutrient DRI values included in this analysis are provided in Appendix A.

#### 2.3.2. Child Anthropometric Assessment

At baseline, child weight was measured to the nearest 0.1 kg using a portable hanging spring scale that was calibrated before each measurement. Recumbent length was measured to the nearest 0.1 cm using a length board with plastic head and footboards and a vinyl measuring surface, following standard techniques [29]. Mothers assisted with taking length measurements, as needed. Using the WHO Anthro Survey Analyzer tool, child length and weight measurements were converted into length for age (LAZ), weight for age (WAZ), and weight for height (WLZ) z-scores [30]. At the time of measurement, children were classified as malnourished or healthy. In accordance with the study’s ethical research approvals, a child was considered malnourished and excluded from the study if his/her WLZ was <−2 SD below the WHO Child Growth Standards median, confirmed through printed and laminated WHO growth standards charts for girls and boys [9]. Excluded children were referred to the local malnutrition program (PROGRESAN) for treatment. All other children were classified as healthy.

#### 2.3.3. Child Food Frequency Questionnaire

At baseline, child feeding practices including human milk and complementary feeding questions were asked. The child food frequency questionnaire (FFQ) was not designed to estimate dietary intake, but rather to assess the frequency of consumption of the following 17 predefined foods/food groups: infant formula, cow’s milk, yogurt, fruit or natural juice, vegetables, chicken, red meat, fish, eggs, beans, cheese, soup or broth, brown rice or rice bran, rice or other cereals, cornmeal/atol, Incaparina, and coffee. The frequency of food intake was reported by mothers and represented in the following five categories: more than daily, daily, between once a day and once a week, previously given but not currently, and never given. Since breast and complementary feeding questions were not asked at endline, endline breastfeeding status was assumed to be the same as that reported at baseline.

### 2.4. Dietary Diversity Scores

All foods and beverages reported in the baseline and endline 24 h dietary recalls were coded into individual food items/ingredients, following the guidelines for the Minimum Dietary Diversity for Women (MDD-W), a diet diversity indicator developed by the Food and Agriculture Organization of the United Nations (FAO) [31]. The food classifications can be found in Appendix A. Dishes were coded into individual ingredients, when possible. Sugar was not counted as an individual food item, and the contents of sugar-sweetened beverages (SSBs) were not separated out by their ingredients but were instead coded as a unit. This included coffee with sugar, horchata, hot chocolate, lemon/pineapple juice with sugar water, instant juice powders, and Coca Cola/soda, among other SSBs. Mixed dishes were classified within one or more food groups, as appropriate and as described in the MDD-W guidelines [31].

For analyzed mothers, we calculated the MDD-W, which was validated for non-pregnant women of reproductive age (15–49 years old), to estimate dietary micro-nutrient adequacy for 10 predefined food groups. To calculate the MDD-W scores, foods reported in the 24 h recalls were categorized into one of the 10 MDD-W food groups. For each mother, a continuous MDD-W score was calculated at baseline and endline, and the two scores were then averaged. Then, per the indicator definition, women were dichotomously classified as achieving MDD-W if they consumed foods in ≥5 of the 10 food groups, or not achieving MDD-W if they consumed <5 of the 10 food groups [31].

For analyzed children, we calculated the minimum dietary diversity (MDD), which was developed by the World Health Organization (WHO) to estimate the previous day’s dietary micro-nutrient adequacy for 8 predefined food groups [32]. For each child, a continuous MDD score was calculated at baseline and endline, and the two scores were then averaged. Then, per the indicator definition, children were dichotomously classified as achieving MDD if they consumed foods in ≥5 of the 8 food groups or not achieving MDD if they consumed <5 of the 8 food groups [32].

### 2.5. Statistical Analysis

All data were cleaned and analyzed in SAS Studio (3.8), and visualizations were created in R (4.2.2). Statistical differences in paired dichotomous baseline and endline dietary diversity scores were assessed via McNemar’s exact chi-square test for mothers and children [33]. Due to the small sample size, outliers were not removed from analysis. Statistical significance was set at *p* < 0.05.

## 3. Results

### 3.1. Characteristics of Study Group

In total, 30 households consented (30 mothers and 30 children). Three children did not meet the study criterion of WLZ ≥ −2 SD and were excluded from analysis. Of the 30 mothers and 27 healthy children, 27 (90%) mothers and 23 (85%) children had complete dietary recalls at both baseline and endline and were included in analyses.

Table 1 shows the baseline (enrollment/week 0) characteristics of the 30 consented households and the analyzed mothers (*n* = 27) and children (*n* = 23). At baseline, on average, five members lived in participating households (SD ± 1.6). Fourteen households (47%) had one child under the age of 5 years, and sixteen (53%) had more than one child under the age of 5 years. Most participants’ homes (70%) were made of cement block, followed by wood (17%) and sheet metal (13%), and almost all (93%) households reported having an outhouse with walls. The average maternal baseline age was 25 years old (SD ± 6), and most mothers were literate (89%). Two mothers (7%) had no formal education, while the majority (59%) had some or complete primary education. Nearly all mothers (93%) reported that they were not employed outside of the household. Among the children, the baseline average age was 12 months old (SD ± 5), and just over half were males (57%). The majority of children were delivered vaginally (61%) and were breastfeeding at baseline (70%).

### 3.2. Assessment of Nutritional Profiles

Median maternal and child nutrient intakes for total energy (calories) and select micro- and macro-nutrients from the baseline and endline dietary recalls before and without dietary intervention are shown in Table 2 along with the percent of the nutrient DRI fulfilled from the median value. The median total calorie intake per day was 1759.5 kcal for lactating mothers, 1374.9 kcal for non-lactating mothers, 582.2 kcal for children 6–12 months old, and 495 kcal for children > 12 months old. A table with additional measures of central tendency can be found in Appendix A. Additional figures of maternal and child average percent DRIs achieved and average calorie intakes modelled with and without the dietary addition of rice bran and/or black beans can be found in Appendix A.

#### 3.2.1. Assessment of Maternal Nutritional Profiles

Analysis of the self-reported regular maternal diets (*n* = 27; Table 3), averaged from the baseline and endline 24 h dietary recalls before and without dietary intervention, shows that all mothers met at least 50% of the DRIs for one macro-nutrient (carbohydrates) and one micro-nutrient (phosphorous). For macro-nutrients, the majority of mothers met at least half of the DRIs for fiber (90%) and protein (96%). Additionally, most mothers consumed at least 50% the DRIs for niacin (85%), calcium (85%), riboflavin (85%), iron (78%), and vitamins E (63%) and B6 (63%). Less than half of mothers met at least 50% the DRIs for potassium (48%), magnesium (37%), choline (26%), zinc (26%), and folate (11%).

Simulating the nutritional contributions of daily intake of 10 g of heat-stabilized rice bran, when added to the regular diet prior to and without intervention, revealed that all mothers (100%) would have met at least half of the DRIs for fiber, niacin, and potassium. The dietary addition of 10 g of rice bran also slightly increased the percentage of mothers consuming at least half of the DRI for riboflavin from 85% to 89%. Doubling rice bran intake to 20 g per day further led to all women meeting at least 50% the DRIs for magnesium and thiamin. In this simulation, improvements were also seen in the percentage of mothers meeting at least half of the DRIs for iron and vitamin E, both of which increased to 82%. Simulation modelling the nutritional contributions with the addition of 100 g of whole, cooked black beans only daily to the regular diet resulted in all mothers meeting at least 50% of the DRI for one additional nutrient: protein. However, simulating the contributions of consuming both 10 g of rice bran and 100 g of cooked black beans daily resulted in all mothers meeting at least half the DRIs for six additional nutrients from those in the regular diet: fiber, niacin, potassium, magnesium, thiamin, and protein. The addition of black beans also increased the percentage of women meeting at least half of the DRIs for folate from 11% to 48% and zinc from 26% to 48%. The results for the simulation models including 20 g of rice bran and 100 g of black beans daily did not lead to all mothers achieving at least half the DRIs for any additional nutrients.

#### 3.2.2. Assessment of Child Nutritional Profiles

Average child baseline weight was 8.6 kg (SD ± 1.4), recumbent length was 73.1 cm (SD ± 6.4), and anthropometric z-scores were weight for age (WAZ) −0.6 ± 0.7, weight for length (WLZ) −0.4 ± 0.8, and length for age (LAZ) −0.5 ± 0.9 (Table 1). Examination of anthropometry according to child age revealed that older children had a slightly lower LAZ compared to those closer to 6 months old (Appendix A).

Analysis of the child diets (*n* = 23; Table 3), averaged from the baseline and endline 24 h dietary recalls before and without dietary intervention, shows that all children met at least 50% of the DRI for one nutrient: protein. For the micro-nutrients, over 80% of all children met at least half of the DRIs for phosphorous (87%), niacin (91%), riboflavin (91%), vitamin B6 (96%), and vitamin B12 (83%). Relatively low percentages of children met at least half of the DRIs for calcium (39%), iron (30%), and folate (35%). Only 30% of children met at least half of the iron DRI.

Modeling the nutritional contributions of the daily intake of 3 g of rice bran added to the regular diet resulted in a scenario where all children (100%) would met at least half of the DRIs for phosphorous, niacin, and magnesium. Improvements from this simulation model were also seen in the percentage of children meeting at least half of the DRIs for carbohydrates, potassium, vitamin E, and choline, which increased to 57%, 52%, 83%, and 52%, respectively. Doubling rice bran intake to 6 g per day further led to all children meeting at least 50% the DRI for thiamin. Simulation modeling of the nutritional contributions with the addition of 10 g of whole, cooked black beans only daily to the regular diet increased the percentage of children achieving at least half of the folate DRI from 35% to 61%. The simulation model including both 3 g of rice bran and 10 g of black beans resulted in all children meeting at least half of the DRIs for phosphorous, niacin, magnesium, and thiamin. No child > 12 months old (*n* = 10) met at least half of the DRI for fiber through their regular diet or any of the dietary simulation models.

### 3.3. Assessment of Child Complementary Feeding Practices

At baseline, of the 27 children considered healthy, 16 (59%, 8 males and 8 females) were being breastfed (Table 1). Among the 13 male children, most (62%) were first introduced to liquids other than breastmilk, including water, at 6 or 8 months of age, while most females (70%) had their first introduction at less than 6 months of age. Most males (69%) and females (60%) were first introduced to solids or semi-solids at 6 months of age (Appendix A).

Males had a higher intake in 15 of the 17 FFQ groups (Appendix A). Females only surpassed males in the frequency of consumption of two food groups: formula and cow’s milk. The child FFQ results showed that most children, regardless of sex, consumed beans (100% males; 75% females), cornmeal/atol (100% males; 75% females), fruit/natural juice (100% males; 75% females), and vegetables (93% males; 83% females) at least once per day or week (Figure 2). Among the sugar-sweetened beverages (SSBs), 80% of males and 50% of females consumed coffee with sugar at least weekly. Of the seven FFQ animal-source foods (ASFs), few male and female children consumed cow’s milk (0% males; 8% females) or yogurt (27% males; 0% females). Male children consumed the other ASFs more frequently than their female peers did, including chicken (87%), fish (87%), and red meat (80%); these were typically consumed by males at least once per day or week. For cheese, most males (67%) also consumed this ASF at least once per week or per day, compared to the 8% of females. The FFQ figure in Appendix A shows these results by child age. Slight differences are also evident in ASF and coffee intake between children aged 6–12 months old and those aged >12 months, with mothers of children aged >12 months reporting more frequent consumption.

### 3.4. Assessment of Dietary Intake by Food Groups

From the maternal and child dietary recalls at baseline and endline, 93 unique foods items were identified and coded. Appendix A lists all food items, as well as their data source and food group classifications. Of the 93 food items, 8 were classified into two food groups and 2 (pizza and chow mein) were classified into four separate food groups, resulting in a total of 107 food classifications. Of these, 16 were classified as grains, 2 as pulses (i.e., beans in this study), 4 as dairy, 14 as meat, 1 as eggs, 2 as dark-green leafy vegetables, 5 as vitamin A-rich fruits or vegetables, 10 as other vegetables, and 1 as other fruit. The remaining 52 (49%) food items were categorized as other, which included 25 condiments, 13 sugar-sweetened beverages, 8 fried/salty items, 3 sweets/desserts, 2 other oils/fats, and water. The other food group did not contribute to achieving minimum dietary diversity. No foods were classified in the nuts or seeds category.

Table 4 shows the percentage of mothers and children that consumed each food group, as well as their dietary diversity score at baseline, at endline, and when averaged. At both baseline and endline, all mothers reported consuming at least one food item within the grains food group. On average, most mothers reported consuming at least one food item from the following food groups: meat (83%), eggs (67%), vitamin A-rich fruits or vegetables (76%), and other vegetables (85%). Few mothers consumed, on average, foods from the dark-green leafy vegetables (7%) and other fruits (2%) categories. At baseline, 18 (67%) of the 27 mothers consumed at least 5 of the 10 MDD-W food groups compared to a slightly lower 16 mothers (59%) at endline. This resulted in an average of 63% of mothers achieving the MDD-W. On average, pulses/beans (28%) and dairy (15%) were the least consumed food groups among children in this study. At baseline, 15 (65%) of the 23 healthy children consumed at least five of the eight MDD food groups compared to a slightly higher 17 children (74%) at endline. This resulted in an average of 16 (70%) of children achieving the MDD between the two recalls. McNemar’s exact chi-square tests did not show statistically significant differences between MDD scores at baseline and endline for mothers or children (Figure 3). Most mothers (*n* = 10, 37%) and children (*n* = 9, 39%) achieved the MDD-W/MDD at both baseline and endline. However, eight mothers (30%) achieved the MDD-W at baseline but not endline, while eight children (35%) did not achieve the MDD at baseline but did at endline.

Individual food items reported in the dietary recalls and consumed on average by at least 10% of all mothers and/or children are shown in Table 5 and split by dietary diversity food group. On average, among the grains, most mothers (96%) and children (80%) consumed corn tortillas. On average, less than a quarter of mothers (24%) and children (22%) reported consuming whole, cooked black beans. Boiled chicken (not fried) was the most consumed meat item, with an average consumption of 50% among mothers and 30% among children. At baseline and endline, over half of all mothers and children reported consuming onions and tomatoes in the dietary recalls, as well as coffee with brown sugar.

## 4. Discussion

This was the first household dietary feasibility study with heat-stabilized rice bran supplementation in rural southwest Guatemala that monitored how rice bran was integrated into beverages and meals for the whole family. Records collected over 3 months of supplementation suggest high compliance, acceptability, and tolerability by mothers and children. This study follows multiple previous in vivo studies that have investigated the dietary and nutritional impacts of rice bran [14,34,35]. Prior to the rice bran dietary intervention, mothers met at least half of the DRIs for only two nutrients (carbohydrates and phosphorous) and children met at least half the DRI for only one nutrient (protein). Our findings demonstrate inadequacies of the traditional diets and the potential for household dietary supplementation of heat-stabilized rice bran coupled with whole, cooked black beans to contribute to the achievement of at least half the DRIs for an additional six nutrients among mothers (fiber, magnesium, niacin, potassium, protein, and thiamin) and four nutrients among children (magnesium, niacin, phosphorous, and thiamin). The daily addition of whole, cooked black beans (100 g for mothers and 10 g for children) allowed for at least 50% of the DRIs for these nutrients to be met when coupled with a smaller amount of rice bran (10 g for mothers and 3 g for children). As black beans are locally available and already consumed in the region, their dietary addition with rice bran could further enhance nutritional status and meet nutrient requirements. During the allocation of the bi-weekly heat-stabilized rice bran packets, households were informally asked about which foods they consumed with the rice bran. Households commonly reported incorporating rice bran into cornmeal/atol, soft drinks, eggs with or without tomatoes, beans with or without broth, and soups. To a lesser extent, households reported mixing rice bran in chicken in mole sauce, chow mein, noodle- or rice-based dishes, tomato sauce, and tamales, among others. The observed simulation models with rice bran (and black beans) applied in this study have practical, low-cost implications to sustainably address food and nutritional security.

Our study showed relatively low dietary fiber intake prior to the addition of rice bran. A study of eight urban Latin American countries showed a fiber intake of 11.2 g/day in the lowest quintile and 22.82 g/day in the highest [36]. Another study in rural Ecuador reported a median fiber intake of 7.7 g/day [37]. Both studies demonstrate participants in the lowest quintile meeting <50% of the fiber DRI for women of reproductive age. Few similar studies have been conducted among children under two years old. An NHANES study found that children 2 years of age consume, on average, 10–18 g/day of fiber, which is 5–13 g/day more than children included in this study [38]. Another NHANES study reported average child fiber intakes to be 10 g/day (or 53% of the fiber DRI) for children 2–4 years old [39]. Our simulation models showed that fiber intake improved in mothers, while fiber intake among children remained low and warrants further attention. The important implications for dietary rice bran prebiotic fibers extends to undernourished children, especially those suffering from diarrheal disease or subclinical inflammatory conditions such as environmental enteric dysfunction (EED), which is characterized by gut dysbiosis and intestinal permeability [20,40]. Dietary soluble and insoluble fibers contained in rice bran and beans have anti-inflammatory and chronic-disease-fighting properties while also supporting gut health [18,41,42].

Rice bran simulation models also estimated nutritional improvements among both mothers and children for magnesium, niacin, and thiamin. However, regular diets without dietary intervention showed low intake for these micro-nutrients. Magnesium is an abundant cation in the body and a cofactor for hundreds of enzymes and enzymatic reactions. Magnesium deficiency is associated with an increased risk of psychiatric, cardiovascular, and gastrointestinal issues, among others, and chronic deficiency can increase the risk of chronic disease, such as hypertension and diabetes [43,44]. Niacin is among the water-soluble B vitamins and comes from exogenous and endogenous sources. Niacin is needed to form nicotinamide adenine dinucleotide (NAD^+^), an essential enzyme influencing energy metabolism, DNA repair, and immune cell function, among others [45]. Emerging evidence supports that higher levels of serum niacin may increase the expression of anti-inflammatory cytokines while decreasing that of pro inflammatory cytokines [46]. This is relevant for overnutrition conditions related to chronic systemic inflammation and adipose tissue releasing pro-inflammatory cytokines, which contribute to obesity-related metabolic complications [47]. Thiamin, an essential micro-nutrient, is a key cofactor in glucose metabolism; therefore, thiamin deficiency alters energy metabolism. During pregnancy, lactation, and neonatal periods, growing fetuses and infants are especially vulnerable to thiamin deficiency [48]. Thiamin deficiency during pregnancy increases the risk of neurological damage to the infant and among pregnant diabetic women; research shows that thiamin supplementation could reduce the risk of infant low birth weight [49]. Thiamin deficiency can also lead to an array of other issues related to cognition and neurodegeneration, memory, motor skills, or heart failure [50]. Rice bran is a promising source of water-soluble B vitamins, including thiamin and niacin [51]. A recent literature review showed that, among higher-income countries, thiamin deficiency can manifest among communities with high white rather than brown rice consumption [52]. The natural form of vitamin E isoforms found in rice bran also warrants attention for having increased bioavailability and biological value to child development and chronic disease prevention [53,54]. Rice bran-derived lipid-soluble vitamin E and its isomers have been extensively investigated for health properties across the lifespan [55,56].

Notably, the majority (over 60%) of mothers and children in this pilot study met the dietary diversity threshold. The relatively high dietary diversity observed herein did apply to the consumption of different food groups, but there was limited variation in food items consumed. For Guatemala, a study showed that approximately 50% of women achieved the MDD-W and with relatively high meat and low dairy intake, which are consistent with our findings [57].

This pilot study revealed unique perspectives to capture household dietary patterns for healthy children in southwest Guatemala, as prior studies focus on dietary interventions among undernourished individuals [58]. The absence of attention paid to healthy body types and the limited co-evaluation of mother–child pairs as household-scale research across LMIC regions is concerning. Food diversity and whole grain intake should be considered in future studies evaluating dietary interventions targeting mother–child pairs. Findings from these dietary recalls identified some consumption variation from baseline and endline averages between food items and groups, which could be related to seasonality, food availability, and cost.

This study’s strengths include the novel approach for a household-level dietary supplement intervention whereby rice bran was incorporated into meals, dishes, and beverages consumed by the whole family, and that the field work was administered by a Project Coordinator who is a native Spanish speaker and resides in the study region. All dietary information was entered as accurately as possible by either taking photos of packaging nutritional information or consulting our local partners, when appropriate. Study limitations include the overall small sample size, not knowing the exact daily consumption quantity of rice bran by the participants, not quantifying breastmilk intake, conducting only one 24 h dietary recall per participant at baseline and endline, and innate limitations to reporting nutritional intake. The lack of a control, non-intervention group is a significant limitation. While the sample size of 30 mother–child pairs was appropriate for this pilot feasibility study, it is analytically weak for making and inferring definitive conclusions. The timing of dietary data collection (e.g., time since last paycheck) could also be considered a limitation when evaluating a small sample size. An endline child complementary feeding questionnaire should also be administered to capture potential changes in food consumption and breastfeeding status. Additionally, we were unable to precisely capture the quantity of some foods consumed. While we recorded how much rice bran each household was provided, we can only speculate as to how it influenced the quantity of other foods that individuals consumed daily. Food classification systems also carry limitations, including the potential to overreport achieving favorable results from dietary recall analyses [59]. Despite these limitations, the high compliance confirmed the feasibility of providing rice bran to households in Guatemala. This pilot study also involved the collection of dried blood spots (DBS) at baseline, midline, and endline for all mother–child pairs, and future analysis of the dietary recall results presented herein will be integrated into DBS metabolomics.

## 5. Conclusions

This pilot study demonstrated the cultural acceptance and tolerability of rice bran incorporated into rural Guatemalan household meals, snacks, and beverages, which led to improvements in meeting the daily dietary requirements for magnesium, niacin, phosphorous, potassium, protein, and thiamin among the mothers and their children. Rice bran consumption led to improvements in fiber intake among mothers but not children. These findings for rice bran-derived micro-nutrients alongside prebiotic fibers have critical implications for other LMIC settings that consume white rice as the major staple food and calorie source. While rice bran is globally produced in large quantities, it is often discarded or used in animal feed. Efforts to increase the availability of rice bran through local manufacturing offer a unique opportunity to harness rice bran’s nutritionally-dense properties in Guatemala. This pilot study lays the foundations for future Guatemala-based research utilizing rice bran as a novel food ingredient in the household with the potential to improve nutritional adequacy among mothers and children.

## Figures and Tables

**Figure 1 nutrients-16-00460-f001:**
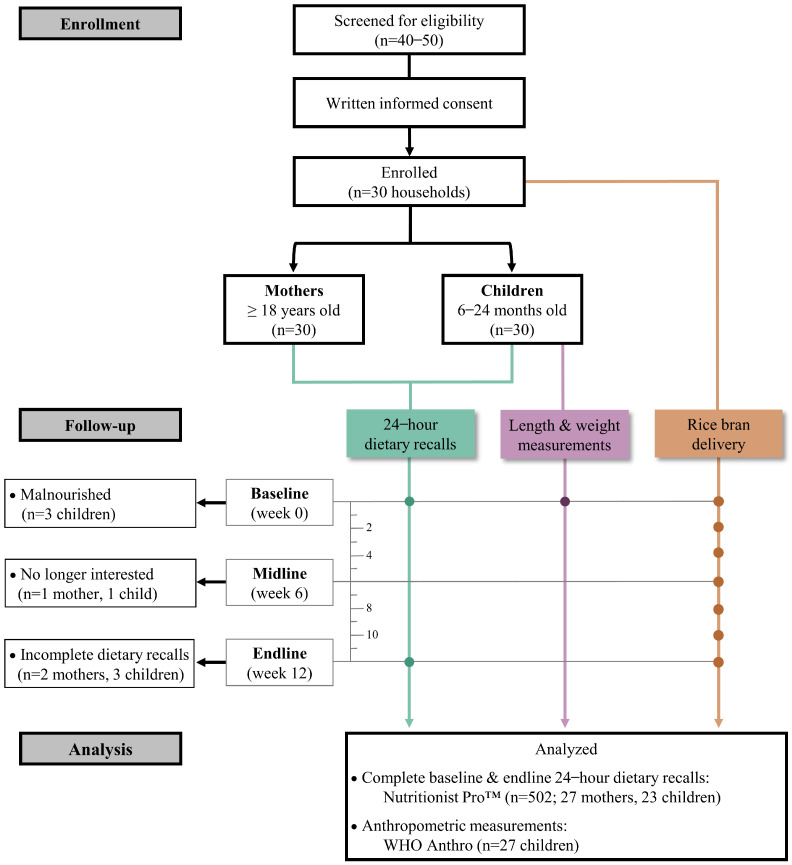
Pilot rice bran household acceptability study, CONSORT. Of the 30 households that consented, 27 mothers and 23 children participated in all 12 weeks of the study and with complete baseline and endline dietary recalls.

**Figure 2 nutrients-16-00460-f002:**
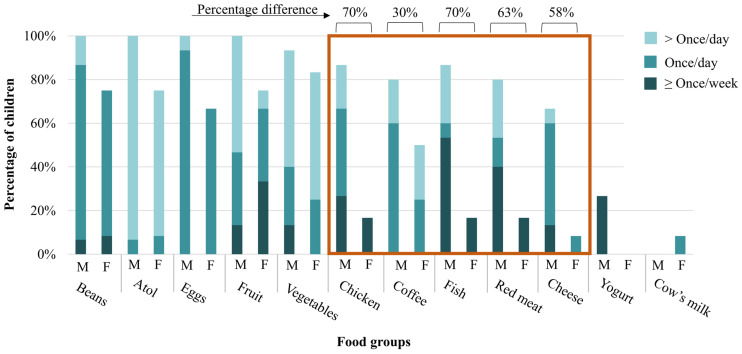
Healthy child (*n* = 27) baseline food frequency questionnaire (FFQ) results for 12 of the 17 food groups included in the FFQ split by sex (M = male/F = female). The fruit food group represents fruit or natural juices. The orange box highlights animal-source foods and coffee, which showed substantial percentage differences in consumption between male and female children. Bars are colored by the percentage of children consuming each food group at least once per week (darkest color), once per day (middle color), and more than once per day (lightest color).

**Figure 3 nutrients-16-00460-f003:**
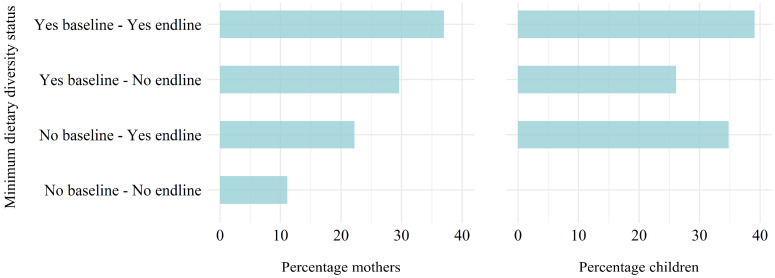
Percentage of mothers (*n* = 27, (**left**)) and children (*n* = 23, (**right**)) meeting the minimum dietary diversity (MDD-W or MDD) requirements from baseline to endline 24 h dietary recalls. The y axis indicates if the mothers or children that had the MDD at both baseline and endline (Yes–Yes), had the MDD at baseline but not at endline (Yes–No), did not have the MDD at baseline but did at endline (No–Yes), or did not have the MDD at either baseline or endline (No–No). McNemar’s exact chi-square test *p*-value = 0.79 for children and for mothers, which suggests no improvement in dietary diversity for either mothers or children from before to after dietary intervention with heat-stabilized rice bran.

**Table 1 nutrients-16-00460-t001:** Household and participant baseline characteristics.

**Households** **(*n* = 30)**	Number of members	5.4 ± 1.6
	Number of children < 5 years old	
	1 child	14 (47)
	>1 child	16 (53)
	Housing type	
	Cement block	21 (70)
	Wood	5 (17)
	Sheet metal	4 (13)
	Drinking water source	
	Bottled	15 (50)
	Piped	7 (23)
	Home well	7 (23)
	Water tank	1 (3)
	Food security	
	Always	28 (63)
	Sometimes/Never	2 (7)
	Sanitation system	
	Outhouse with walls	28 (93)
	Outhouse without walls	1 (3)
	Indoor toilet	1 (3)
**Mothers** **(*n* = 27)**	Age in years	25 ± 6
	18–30 years old	21 (78)
	31–40 years old	6 (22)
	Lactating	20 (74)
	Literacy	
	Literate	24 (89)
	Illiterate	3 (11)
	Highest level of education completed	
	No formal education	2 (7)
	Some primary	10 (37)
	Completed primary	6 (22)
	Some/Completed secondary	7 (26)
	Tech training/University	2 (7)
	Married/In a union	24 (89)
	Work status	
	Full-time	1 (4)
	Part-time	1 (4)
	Does not work outside of the home	25 (93)
**Children** **(*n* = 23)**	Age in months	12 ± 5
	6–12 months old	13 (57)
	13–24 months old	10 (43)
	Sex	
	Male	13 (57)
	Female	10 (43)
	Delivery method	
	Vaginal	14 (61)
	C-section	9 (39)
	Currently breastfeeding at baseline	
	Yes	16 (70)
	No	7 (30)
	Anthropometry	
	Weight, kg	8.6 ± 1.4
	Height, cm	73.1 ± 6.4
	Weight-for-age z-score (WAZ)	−0.6 ± 0.7
	Weight-for-length z-score (WLZ)	−0.4 ± 0.8
	Length-for-age z-score (LAZ)	−0.5 ± 0.9

Values are shown as mean ± standard deviation or *n* (%).

**Table 2 nutrients-16-00460-t002:** Dietary intake measures of central tendency for mothers and children without dietary intervention.

Nutrient	Units	Mothers							
		Lactating (*n* = 20)				Not Lactating (*n* = 7) ^1^			
		Q1	Median	Q3	Median %DRI	Q1	Median	Q3	Median %DRI
**Calories**									
Calories	kcal	1545.7	1759.5	2299.3		1050.1	1374.9	2220.7	
**Macro-nutrients**									
Carbohydrates	g	219.9	251.5	324.4	120%	148.2	202.4	292.1	156%
Added sugar	g	9.5	20.6	31.8		5.3	23.7	35.9	
Cholesterol	mg	192.5	335.6	436.2		119.4	201.6	363.8	
Total fat	g	40.4	49.1	74		26.7	37.7	84.6	
Total dietary fiber	g	15.8	19.4	24.9	67%	12.6	14.1	18.9	56%
Protein	g	54.3	70.1	82.7	99%	34.4	47.8	60	104%
**Micro-nutrients**									
Biotin	mcg	1.9	4.5	6.9	13%	1.1	1.5	2.1	5%
Calcium	mg	521.7	673.7	837.4	67%	433	504.8	537.6	50%
Choline	mg	120.1	211.2	280.7	38%	87.1	139.5	261.9	33%
Total folate	mcg	85.3	145.4	194.7	29%	56.4	63.4	71	16%
Iron	mg	7.5	9.7	13.6	108%	4.5	4.8	9.3	27%
Magnesium ^2^	mg	108.1	139.1	182.1	44%	81.5	90.3	166.1	29%
Manganese	mg	0.4	0.8	1.4	29%	0.2	0.3	2.2	16%
Niacin	mg	15.8	19.6	23.4	115%	8.5	10.9	11.5	78%
Pantothenic acid	mg	1.1	1.9	2.6	28%	0.4	1.4	2	27%
Phosphorus	mg	923.6	1047.3	1283.5	150%	539.5	884.1	950.5	126%
Potassium	mg	1204	1611.2	1948.8	58%	827.3	988.6	1513.9	38%
Riboflavin	mg	0.9	1.1	1.3	67%	0.4	0.8	0.9	74%
Selenium	mcg	37.3	53.4	89.5	76%	19.3	33.8	71.6	62%
Thiamin	mg	0.7	0.9	1.1	66%	0.6	0.6	0.9	59%
Vitamin B6	mg	0.9	1.1	1.5	56%	0.5	0.8	1.2	62%
Vitamin B12	mcg	1.1	1.6	2.2	57%	0.5	1.2	1.6	50%
Vitamin C	mg	22	36	68.3	30%	10.3	15.5	39.3	21%
Vitamin D	mcg	1.1	2.4	3.2	16%	0.6	1.2	2.1	8%
Vitamin E	mg	5.7	11.7	21.6	62%	5.9	12.7	38.2	85%
Vitamin K	mcg	11.6	18.7	54.4	21%	5.1	7.2	8.9	8%
Zinc	mg	4.5	5.4	6.3	45%	2.7	3.1	3.4	39%
**Nutrient**	**Units**	**Children**							
		**6–12 months old (*n* = 13)**				**>12 months old (*n* = 10)**			
		**Q1**	**Median**	**Q3**	**Median %DRI**	**Q1**	**Median**	**Q3**	**Median %DRI**
**Calories**									
Calories	kcal	425.8	582.2	680.4		452.3	495	655.9	
**Macro-nutrients**									
Carbohydrates	g	38.2	74.9	96.9	79%	40	62.9	72.2	48%
Added sugar	g	0	2.6	6		1.3	8.7	14.4	
Cholesterol	mg	59.8	111.1	154.9		106.2	128.9	179.6	
Total fat	g	17.4	18.7	26.5	62%	17.4	19.9	25	
Total dietary fiber	g	3.4	6.5	7.8		3.4	4	4.4	21%
Protein	g	14.2	23.2	33.1	211%	14.2	22.5	26.2	173%
**Micro-nutrients**									
Biotin	mcg	0.3	1.2	2.9	20%	0.6	1.7	3.4	22%
Calcium	mg	121.6	150	221	58%	143.6	153.4	186	22%
Choline	mg	46.1	81.2	97.2	54%	68.4	84.8	112.4	42%
Total folate	mcg	29	41.9	64.6	52%	32.3	41.5	72.4	28%
Iron	mg	1.8	3.6	4.7	33%	3.1	3.4	3.9	48%
Magnesium ^2^	mg	30.7	61.2	89.1	82%	50.7	59.7	73.7	75%
Manganese	mg	0.1	0.2	0.6	34%	0.2	0.3	0.4	25%
Niacin	mg	3.1	6.9	8.8	171%	3.6	5.7	7.1	94%
Pantothenic acid	mg	0.4	0.9	1.6	52%	0.5	0.7	0.9	36%
Phosphorus	mg	226.5	369.1	432.2	134%	218.6	330.7	374.2	72%
Potassium	mg	409.8	659.8	904.8	77%	496.6	639	661	32%
Riboflavin	mg	0.3	0.3	0.6	78%	0.3	0.4	0.4	71%
Selenium	mcg	13.3	26	45.3	130%	12.9	19.7	29.5	99%
Thiamin	mg	0.2	0.2	0.4	74%	0.2	0.3	0.3	51%
Vitamin B6	mg	0.3	0.5	0.8	168%	0.4	0.4	0.5	89%
Vitamin B12	mcg	0.4	0.6	0.9	113%	0.5	0.7	0.9	79%
Vitamin C	mg	10	27.8	39.9	56%	5	14.1	25.3	94%
Vitamin D	mcg	0.6	1	2.2	10%	0.6	1	1.5	7%
Vitamin E	mg	2.7	4.3	8.2	86%	2.5	5.5	8.5	92%
Vitamin K	mcg	3.7	7.7	10.1	307%	2.6	6.7	21.5	22%
Zinc	mg	1.3	2.3	2.7	76%	1.6	1.6	1.8	55%

^1^ For women who reported to not be lactating, the DRIs for women 18–30 years of age were used to calculate the median % DRI values. ^2^ For manganese, the DRI for women 18–30 years old was used to compute the median % DRI.

**Table 3 nutrients-16-00460-t003:** Percentage of mothers and children achieving ≥50% of the nutrient dietary reference intakes (DRIs), ordered by the largest % of mothers meeting at least 50% of the DRIs.

	Mothers (*n* = 27)						Children (*n* = 23)					
Nutrient ^1^	Regular Diet ^4^	+10 g RB	+20 g RB	+100 g Beans	+10 g RB & 100 g Beans	+20 g RB & 100 g Beans	Regular Diet	+3 g RB	+6 g RB	+10 g Beans	+3 g RB & 10 g Beans	+6 g RB & 10 g Beans
**Contributions from heat-stabilized rice bran and/or cooked, whole black beans**												
Carbohydrates ^2,3^	**100**	100	100	100	100	100	47.8	56.5	65.2	56.5	65.2	65.2
Phosphorous ^2,3^	**100**	100	100	100	100	100	87.0	**100**	100	91.3	**100**	100
Dietary fiber ^2,3,^*	88.9	**100**	100	92.6	**100**	100	0.0	0.0	0.0	0.0	0.0	0.0
Niacin ^2,3^	85.2	**100**	100	88.9	**100**	100	91.3	**100**	100	91.3	**100**	100
Potassium ^2,3^	48.2	**100**	100	77.8	**100**	100	43.5	52.2	52.2	52.2	52.2	56.5
Magnesium ^2,3^	37.0	88.9	**100**	63.0	**100**	100	73.9	**100**	100	78.3	**100**	100
Thiamin ^2,3^	74.1	96.3	**100**	92.6	**100**	100	69.6	91.3	**100**	78.3	**100**	100
Protein ^2,3^	96.3	96.3	96.3	**100**	**100**	100	**100**	100	100	100	100	100
Riboflavin ^2,3^	85.2	88.9	88.9	88.9	88.9	88.9	91.3	91.3	95.7	91.3	91.3	95.7
Calcium ^2,3^	85.2	85.2	85.2	88.9	88.9	88.9	39.1	39.1	39.1	39.1	39.1	39.1
Iron ^2,3^	77.8	77.8	81.5	81.5	85.2	85.2	30.4	34.8	47.8	30.4	47.8	47.8
Vitamin E ^2^	63.0	70.4	81.5	63.0	70.4	81.5	73.9	82.6	82.6	73.9	82.6	82.6
Vitamin B6 ^3^	63.0	63.0	63.0	77.8	77.8	77.8	95.7	95.7	95.7	95.7	95.7	95.7
Sodium ^2,3^	59.3	59.3	59.3	59.3	59.3	59.3	65.2	65.2	65.2	65.2	65.22	65.2
Vitamin C ^3^	29.6	29.6	29.6	29.6	29.6	29.6	60.9	60.9	60.9	60.9	60.9	60.9
Choline ^2^	25.9	25.9	29.6	25.9	25.9	29.2	47.8	52.2	52.2	47.8	52.2	52.2
Zinc ^3^	25.9	25.9	25.9	48.2	48.2	48.2	78.3	78.3	78.3	78.3	78.3	78.3
Folate ^3^	11.1	11.1	11.1	48.2	48.2	48.2	34.8	34.8	34.8	60.9	60.9	60.9
**Other nutrients**												
Selenium	74.1	74.1	74.1	74.1	74.1	74.1	87.0	87.0	87.0	87.0	87.0	87.0
Vitamin B12	66.7	66.7	66.7	66.7	66.7	66.7	82.6	82.6	82.6	82.6	82.6	82.6
Copper	29.6	29.6	29.6	29.6	29.6	29.6	78.3	78.3	78.3	78.3	78.3	78.3
Vitamin K	18.5	18.5	18.5	18.5	18.5	18.5	65.2	65.2	65.2	65.2	65.2	65.2
Pantothenic acid	7.4	7.4	7.4	7.4	7.4	7.4	34.8	34.8	34.8	34.8	34.8	34.8
Biotin	0.0	0.0	0.0	0.0	0.0	0.0	13.0	13.0	13.0	13.0	13.0	13.0
Molybdenum	0.0	0.0	0.0	0.0	0.0	0.0	13.0	13.0	13.0	13.0	13.0	13.0
Vitamin D	0.0	0.0	0.0	0.0	0.0	0.0	8.7	8.7	8.7	8.7	8.7	8.7

^1^ All nutrient values are based on the recommended dietary allowance (RDA) value, if available. Adequate intake (AI) was used when a nutrient RDA was not set. Appendix A includes a list of the nutrients and their reported DRI type (i.e., RDA/AI). ^2^ Heat-stabilized rice bran nutritional contributions. ^3^ Cooked, whole black bean nutritional contributions. ^4^ Percentage of mothers/children meeting dietary requirements from average baseline and endline nutrient values without any dietary intervention. RB = heat-stabilized rice bran; beans = cooked, whole black beans. Values are percentages calculated based on individual status or age group for mothers (one of three groups: lactating, 18–30 years old, or 31–50 years old) and children (one of two groups: 6–12 months or 13 months–3 years old). * As there is no fiber DRI for children < 12 months old, fiber values for children in Table 3 are for those aged 13–24 months only (*n* = 10). Bolded values represent the first time all participating mothers or children met at least 50% of the nutrient DRI.

**Table 4 nutrients-16-00460-t004:** Percentage of mothers (*n* = 27) and children (*n* = 23) consuming foods categorized in the minimum dietary diversity food groups; 10 food groups for women/mothers and 8 food groups for children.

Food Groups ^1^	Percentage Consumption					
	Mothers ^2^ (*n* = 27)			Children ^3^ (*n* = 23)		
	Baseline	Endline	Average	Baseline	Endline	Average
Grains (16)	100	100	100	100	91.3	95.7
Pulses (2)	29.6	37.0	33.3	34.8	21.7	28.3
Nuts and seeds (0)	0	0	0	–	–	–
Dairy (4)	22.2	7.4	14.8	17.4	13.0	15.2
Meat (13)	81.5	85.2	83.3	69.6	78.3	73.9
Eggs (1)	70.4	63.0	66.7	52.2	65.2	58.7
Vitamin A-rich fruits and vegetables (5)	77.8	74.1	75.9	60.9	73.9	67.4
Dark-green leafy vegetables (2)	7.4	7.4	7.4	73.9	82.6	78.3
Other vegetables (10)	81.5	88.9	85.2			
Other fruit (1)	0.0	3.7	1.9			
Breast milk	–	–	–	69.6	69.6	69.6
Minimum dietary diversity	66.7	59.3	63.0	65.2	73.9	69.6

^1^ Number of food items categorized in each food group are in parentheses. ^2^ Maternal dietary diversity scores were calculated following the minimum dietary diversity for women (MDD-W) developed by the Food and Agriculture Organization of the United Nations (FAO) [31]. ^3^ Child dietary diversity scores were calculated following the minimum dietary diversity indicator developed by the World Health Organization (WHO) [32].

**Table 5 nutrients-16-00460-t005:** Percentage of mothers and children reporting consuming select household foods that contribute to the dietary diversity food groups.

Food Items by Food Group ^1^	Percentage Consumption					
	Mothers (*n* = 27)			Children (*n* = 23)		
	Baseline	Endline	Average	Baseline	Endline	Average
**Grains**						
Tortilla	96.3	96.3	96.3	82.6	78.3	80.4
Sweet bread	29.6	25.9	27.8	17.4	8.7	13.0
Pasta/Spaghetti	18.5	18.5	18.5	21.7	13.0	17.4
French bread	18.5	18.5	18.5	26.1	17.4	21.7
Potatoes	11.1	14.8	13.0	17.4	8.7	13.0
**Pulses**						
Black beans (cooked, whole)	29.6	18.5	24.1	30.4	13.0	21.7
Pureed black beans	3.7	18.5	11.1	4.4	8.7	6.5
**Meat**						
Chicken—not fried	48.2	51.9	50.0	26.1	34.8	30.4
Red meat	25.9	33.3	29.6	21.7	26.1	23.9
Fish	25.9	0.0	13.0	26.1	0.0	13.0
Chicken—fried	3.7	7.4	5.6	13.0	13.0	13.0
**Vegetables**						
Onion	74.1	66.7	70.4	69.6	60.9	65.2
Tomatoes (whole)	70.4	55.6	63.0	52.2	60.9	56.5
Carrots	11.1	25.9	18.5	21.7	21.7	21.7
Chayote	7.4	18.5	13.0	8.7	13.0	10.9
**Dairy**						
Cheese	18.5	7.4	13.0	8.7	8.7	8.7
**Other**						
Sugar-sweetened beverages	88.9	100.0	94.4	73.9	78.3	76.1
Coffee with sugar	77.8	96.3	87.0	73.9	52.2	63.0
Coke/Soda	29.6	44.4	37.0	0.0	30.4	15.2
Fruit juice	7.4	7.4	7.4	4.4	21.7	13.0
Cookies	3.7	11.1	7.4	0.0	21.7	10.9

^1^ Food items included in the table were consumed by at least 10% of mothers and/or children. Foods were classified following the minimum dietary diversity for women (MDD-W) indicator guidelines [31].

## Data Availability

Data are available upon request from authors.

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
