# Peer review of "Pilot Study of Heat-Stabilized Rice Bran Acceptability in Households of Rural Southwest Guatemala and Estimates of Fiber, Protein, and Micro-Nutrient Intakes among Mothers and Children"

_nutrients, 2024, doi:10.3390/nu16030460_

Round 1

Reviewer 1 Report

Comments and Suggestions for Authors

The aim of this pilot study was to examine the regular diets with/without dietary rice bran intervention of mother-child pairs in the El Trifinio region of rural southwest Guatemala, and to model nutritional contributions of daily consumption of heat-stabilized rice bran and/or whole, cooked black beans.

The subject discussed is important and interesting, but I would like to draw attention to certain issues:

- what values (DRI) were adopted for the analyzed nutrients?

-why/on what basis was “achieving ≥50% of the dietary reference intake (DRI)” considered as a reference point (table 2)?

- what do regular diet  and regular diet4 mean (table 2 - in the footnote to the table: 4Average baseline and endline nutrient values without any dietary intervention. 5RB = heat-stabilized rice bran; 4beans = cooked black beans)? 

- how the content of nutrients in Rice Bran 300 was determined?

- I miss reference to the possible bioavailability of nutrients from rice bran, which may be limited due to the fiber content; secondly, fat-soluble compounds, such as the mentioned vitamin E, require the presence of fat in the diet, which was omitted in this analysis

- why anthropometric measurements were not included at the end of the intervention?; this would allow for a better assessment of the usefulness of using such a product

I hope that considering the above comments will also allow to improve the conclusions, which should not include references to the literature, but indicate the future direction.

Minor comments:

- add "pilot study" to the title

- some wording needs improvement: Intervention dosing; Child anthropometry; Child food frequency; Maternal and child nutritional profiles; Maternal dietary recall etc.; Percent consumption/mothers...

- percentage - no percent

Author Response

Reviewer #1 comments:

  1. what values (DRI) were adopted for the analyzed nutrients?

Response:

            These values can be found in the Excel supplemental file titled “Supplemental table S2_Diet intake and diversity” under the tab titled “DRIs”. We have added a sentence in the main text to clarify this point and the fact that the DRIs were derived by the Institute of Medicine (IOM) (lines 165-168).

  1. -why/on what basis was “achieving ≥50% of the dietary reference intake (DRI)” considered as a reference point (table 2)?

Response:

            There is limited knowledge of whether mother-child pairs are meeting dietary guidelines in the area.  Ideally 100% of the DRIs would be achieved, yet we have found this is often not the case.

Guatemala is an especially nutritionally vulnerable population. In order to establish realistic nutrient intake goals among this population, we developed the table to see the percentage of mothers and children that would achieve at least 50% of the DRIs with the dietary addition of rice bran and black beans.

Additional support for this response is provided from Analysis of NHANES data that have shown most individuals have nutrient intakes below the recommended levels.

A study by Ahluwalia et al. (2016) showed that less than 1% of children 12-23 months old at least met the AI for fiber and potassium [1]. Other NHANES studies have also shown that women of reproductive age in the US tend to consume nutrients below the recommended values [2]. Furthermore, the DRIs were developed based on the healthy US and Canadian population and are not are not context-specific values [3].

  1. what do regular diet  and regular diet4 mean (table 2 - in the footnote to the table: 4Average baseline and endline nutrient values without any dietary intervention. 5RB = heat-stabilized rice bran; 4beans = cooked black beans)? 

Response:

            The regular diet refers to dietary intakes prior to and without dietary intervention. This was reworded and emphasized for clarity further in the manuscript (e.g., lines 139-140, 165-166, 276, 302).

  1. how the content of nutrients in Rice Bran 300 was determined?

Response:

            RiceBran Technologies (Scottsdale, AZ, USA) now operates under the company name Stabil Nutrition (https://stabilnutrition.com) and performs a standardized product analysis that includes, but is not limited to macro and micronutrients as well as selected phytochemicals (protein, fat, carbohydrates, B and E vitamins, gamma oryzanol, and other phytosterols). Each lot of production (product spec sheet available upon request) is also screened for mycotoxins and heavy metals to ensure product safety. All product received for this study was tested and approved for human consumption as a food ingredient.  The nutritional contents of the heat stabilized rice bran purchased and supplied by RiceBran Technologies is included as the first figure in the Supplemental figures S1 file and referenced in the manuscript (lines 113-115 and 163-164). The supplemental figure has each standard analysis with the appropriate number written next to it in parenthesis.

  1. I miss reference to the possible bioavailability of nutrients from rice bran, which may be limited due to the fiber content; secondly, fat-soluble compounds, such as the mentioned vitamin E, require the presence of fat in the diet, which was omitted in this analysis

Response:

            Thank you for these comments. Bioavailability of rice bran nutrients and key components will be assessed via analysis of dried blood spots that were collected on the mothers and children before, during and after rice bran supplementation. Nutritional analysis of the diet that is described for mothers and children was from dietary food logs and is distinct from establishing enhanced bioavailability of specific components.

To show the presence of fat in the diet, we have added a table (Table 2, pages 9 and 10) to the manuscript indicating measures of central tendency (median, Q1, Q2 and percent DRI from the median value) of calories and select micro and macronutrients for mothers (lactating / non-lactating) and children (6-12months /  > 12 months old). A more complete table has also been included in Supplemental table S2.

  1. why anthropometric measurements were not included at the end of the intervention?; this would allow for a better assessment of the usefulness of using such a product

Response:

            While we do agree that including anthropometric measurements for both the children and mothers to the endline would be interesting in response to the household rice bran consumption, the short duration of the pilot study (3 months) and small sample size were considered limitations for generalizable results interpretations. Dietary rice bran supplementation was also not controlled in this study and differences in compliance to intake doses would be difficult to interpret to changes observed in anthropometry, especially for child length. Based on the compelling outcomes and demonstrated feasibility of this pilot study, we have designed and proposed a randomized controlled trial investigation for 6 months that will include reporting both baseline and endline anthropometric measurements.

  1. I hope that considering the above comments will also allow to improve the conclusions, which should not include references to the literature, but indicate the future direction.

Response:

            The reference was removed from the Conclusions section. The text in the Conclusion was modified to incorporate future directions (lines 531-544).

  1. add "pilot study" to the title

Response:

            The title was changed to include ‘pilot study’ (line 2). The word pilot was also added to the abstract (line 29), introduction (line 83), the material and methods title (line 90), and conclusion (lines 531 and 544).

  1. some wording needs improvement: Intervention dosing; Child anthropometry; Child food frequency; Maternal and child nutritional profiles; Maternal dietary recall etc.; Percent consumption/mothers...

Response:

            Thank you for these comments. The wording that was cited above is commonly used in maternal and child nutrition related scientific literature and was deemed appropriate to the details for this pilot study.

  1. percentage - no percent

Response:

            Thank you. This wording was fixed and highlighted in yellow throughout the manuscript.

References:

  1. Ahluwalia, N.; Herrick, K.A.; Rossen, L.M.; Rhodes, D.; Kit, B.; Moshfegh, A.; Dodd, K.W. Usual Nutrient Intakes of US Infants and Toddlers Generally Meet or Exceed Dietary Reference Intakes: Findings from NHANES 2009–2012. Am J Clin Nutr 2016, 104, 1167–1174, doi:10.3945/ajcn.116.137752.
  2. Devarshi, P.P.; Legette, L.L.; Grant, R.W.; Mitmesser, S.H. Total Estimated Usual Nutrient Intake and Nutrient Status Biomarkers in Women of Childbearing Age and Women of Menopausal Age. Am J Clin Nutr 2021, 113, 1042–1052, doi:10.1093/ajcn/nqaa392.
  3. https://ods.od.nih.gov/HealthInformation/nutrientrecommendations.aspx

Reviewer 2 Report

Comments and Suggestions for Authors

The work presented for review is very interesting and deals with the extremely important topic of prevention. Often, as the authors point out, the studies refer to people who were found to be undernourished, overweight or obese, and the authors assessed children from the so-called mass norm.

Although the work was very interesting, the authors made a few mistakes:

1. no control group

2. why were children assessed in such a wide age range, especially at this age - the average age, according to the authors, was 12 months SD 5 months and the table shows the groups 6-12 months and 13-24 months - this is a very diverse way of feeding, especially since 70% of the children were still breastfed.

If there was more than one child of this age at home, were the tests covered all or only selected people?

3. On what basis did the mother assess what the child ate while she was at work

4. in the results, the authors show a group who were given 100 g of black beans (mothers line 277-286) and 10 g of black beans (children line 308-311), but it is not known whether it was a special diet, as in the case of rice bran, or standard food for this social group.

5 in the discussion section, the authors focused more on re-describing the importance of nutrients than on comparing the obtained results with other studies. No reference to other studies

6. the authors drew far-reaching conclusions based on the results of a small group tested additionally from 1 recorded feeding day. The lack of control over the diet and meals consumed between weeks 0 and 12 raises concerns about the reliability of the research, especially given the relatively low education of the mothers studied.

The work requires supplementing with a comparison group

Author Response

Reviewer #2 comments:

Although the work was very interesting, the authors made a few mistakes:

  1. no control group

Response:

This was intended to be a pilot feasibility study for household supplementation with dietary rice bran. Rice bran is a novel ingredient, and little is known about the cultural acceptability and tolerability when added to the local diet. Follow up studies will use a control supplement product.   We agree that the lack of a control group is a significant study limitation. This was added to the description of limitations to the Discussion section (line 514-515).

  1. why were children assessed in such a wide age range, especially at this age - the average age, according to the authors, was 12 months SD 5 months and the table shows the groups 6-12 months and 13-24 months - this is a very diverse way of feeding, especially since 70% of the children were still breastfed. If there was more than one child of this age at home, were the tests covered all or only selected people?

Response:

      The study design and protocol outlined eligibility criteria for enrollment of 30 households, and to collect data on a mother-child pair. We included only healthy children (not malnourished) between 6-23 months of age to participate in the study because this is the age group with highest rates of chronic malnutrition in Guatemala, and we wanted to assess cultural acceptability of rice bran into the local foods/diet. Notably,  there were no households that had more than one child that fit the age criteria. We were not able to advise a family with a malnourished child to use an untested feeding protocol, for ethical reasons. As a result, we only included children that were not malnourished (as per our definition) in the study.

  1. On what basis did the mother assess what the child ate while she was at work

Response:

            Of the 27 mothers, only 2 (7%) worked outside of the household, one of which worked part-time and the other full-time. All the other participating mothers reported working at home. Table 1 (line 250) was modified to include the distinction between full- and part-time work. While we did not ask the 2 mothers how they assessed what the child ate, it is customary in the area for the mothers to bring young children to work with them. If they did not do this, it was assumed they consulted with the individual providing care when they got back from work. However, we appreciate this comment and will integrate it into future study designs.

  1. in the results, the authors show a group who were given 100 g of black beans (mothers line 277-286) and 10 g of black beans (children line 308-311), but it is not known whether it was a special diet, as in the case of rice bran, or standard food for this social group.

Response:

Black beans are regularly consumed in the region and was not part of the supplementation to the diet. Text clarifying this point was added to the discussion section (lines 435-440).

  1. in the discussion section, the authors focused more on re-describing the importance of nutrients than on comparing the obtained results with other studies. No reference to other studies

Response:

The discussion section was modified to discuss more results from other studies (lines 443-454, 496-503).

  1. the authors drew far-reaching conclusions based on the results of a small group tested additionally from 1 recorded feeding day. The lack of control over the diet and meals consumed between weeks 0 and 12 raises concerns about the reliability of the research, especially given the relatively low education of the mothers studied.

 Response:

            Having one recorded feeding day at baseline and one recorded feeding at endline is included in the manuscript as a limitation (lines 513-514).

We agree that the lack of a control group is a significant study limitation. To further emphasize this point, we removed this from the list of limitations and instead added a separate sentence regarding this point to the Discussion section (line 514-515).

  1. The work requires supplementing with a comparison group

Response:

We agree that the lack of a control group is a significant study limitation. To further emphasize this point, it was added to the description of limitations to the Discussion section (line 514-515).

Reviewer 3 Report

Comments and Suggestions for Authors

The manuscript by Pfluger et al., describes interesting research, but sometimes it is too complicated to follow, especially when it comes to describing the methods and presenting the results.

Even if authors stated in the Discussion that one of the limitation of the present manuscript is the small sample size of mother-child pairs, did the author performed a sample size calculator? If not, why?

In addition, it would have been interesting to know the composition in essential/non essential amino acid, and changes in the gut microbiota.

Author Response

Reviewer #3 comments:

  1. The manuscript by Pfluger et al., describes interesting research, but sometimes it is too complicated to follow, especially when it comes to describing the methods and presenting the results.

Response:

            The methods and results sections were edited in an effort to improve the manuscript language. This included, among others, the following:

  • Changing El Trifinio to Retalhuleu Department throughout the text.
  • Writing more concisely throughout the sections including more extensive revisions to sub-section 2.3. Dietary recalls.
  • Adding definitions, where appropriate (e.g., lines 97-98).

  1. Even if authors stated in the Discussion that one of the limitation of the present manuscript is the small sample size of mother-child pairs, did the author performed a sample size calculator? If not, why?

Response:

This was the first dietary rice bran supplementation study in the rural area of southwest Guatemala, and we had limited information regarding the number of households that would be meeting minimum standards of dietary quality in mothers and healthy children.  Therefore, estimation of a sample size is a challenge. We can now use these pilot data generated in 30 household for performing a power calculation and sample size estimation needed for a randomized control trial.

  1. In addition, it would have been interesting to know the composition in essential/non essential amino acid, and changes in the gut microbiota.

Response:

            Thank you for this comment. Stool sample collection for performing microbiome analysis was not included in this pilot study. We have a grant proposal to expand upon these study results and include the gut microbiome.

Round 2

Reviewer 1 Report

Comments and Suggestions for Authors

Thank you for considering my comments, but I cannot accept yet due to the subheadings. Moreover, the use of words such as "population" or "cohort" is unjustified.

Subtitles in Results should be different from methods subtitles and focus on evaluation/analysis.

Comments on the Quality of English Language

 I cannot accept yet due to the subheadings:

is: Pilot study design and population - should be: Study design and group

Intervention dosing -  Dietary intervention

 Child anthropometry - Assessment of the child's nutritional status

Child food frequency - Child's nutrition assessment

Pilot study cohort - Characteristic of study group

Author Response

  1. Moreover, the use of words such as "population" or "cohort" is unjustified.

Response: Thank you for this comment. We have replaced the words ‘population’ and ‘cohort’ and have highlighted these changes in the manuscript (lines 449, 494, 503, 507).

  1. Subtitles in Results should be different from methods subtitles and focus on evaluation/analysis.

Response:

We made the following changes in the headings in the Materials and methods section:

Pilot study design and population - Study design and group (line 92)

Intervention dosing -  Dietary intervention (line 114)

Data collection (new subheading, line 132)

Child anthropometry – Child anthropometric assessment (line 174)

Child food frequency – Child food frequency questionnaire (line 189)

Dietary diversity – Dietary diversity scores (line 201)

We made the following changes in the headings in the Results section:

Pilot study cohort - Characteristics of study group (line 235)

Maternal and child nutritional profiles – Assessment of nutritional profiles (line 255)

Maternal dietary recalls – Assessment of maternal nutritional profiles (line 263)

Child dietary recalls – Assessment of child nutritional profiles (line 290)

Child complementary feeding – Assessment of child complementary feeding practices (line 330)

Dietary diversity – Assessment of dietary intake by food groups (line 362)

  1. I cannot accept yet due to the subheadings: Pilot study design and population - should be: Study design and group, Intervention dosing -  Dietary intervention, Child anthropometry - Assessment of the child's nutritional status, Child food frequency - Child's nutrition assessment, Pilot study cohort - Characteristic of study group

Response:

The following subheadings were changed, as requested:

Pilot study design and population - Study design and group (line 92)

Intervention dosing -  Dietary intervention (line 114)

Pilot study cohort - Characteristics of study group (line 235)

We put the following under the new subheading Data collection (line 132):

Dietary recalls (line 133)

Child anthropometric assessment (line 174)

Child food frequency questionnaire (line 189)

Child anthropometry - Assessment of the child's nutritional status:

We changed the subheading in the Methods section from Child anthropometry to Child anthropometric assessment (line 174). We placed the anthropometric results under the subheading Assessment of child nutritional profiles (line 290).

Child food frequency - Child's nutrition assessment:

As the subsection details the food frequency questionnaire (FFQ) we feel as though changing this subheading in the methods section may confuse readers in distinguishing between the FFQ and the dietary recalls. We slightly modified these headings. In the Methods section, we changed the subheading Child food frequency to Child food frequency questionnaire (line 189). We placed the anthropometric results under the subheading Assessment of child nutritional profiles (line 290).

Reviewer 2 Report

Comments and Suggestions for Authors

In its current form, the article is suitable for printing

Author Response

Response: Thank you for your time and feedback.